# Characteristics of Surface Electromyograph Activity of Cervical Extensors and Flexors in Nonspecific Neck Pain Patients: A Cross-Sectional Study

**DOI:** 10.3390/medicina58121770

**Published:** 2022-11-30

**Authors:** Ruka Nobe, Hiroyoshi Yajima, Miho Takayama, Nobuari Takakura

**Affiliations:** Department of Acupuncture and Moxibustion, Tokyo Ariake University of Medical and Health Sciences, Tokyo 135-0063, Japan

**Keywords:** nonspecific neck pain, surface electromyogram, craniovertebral angle, cervical vertebrae, neck muscles, electromyography

## Abstract

*Background and Objectives:* We identified typical surface electromyogram (sEMG) activities of the cervical extensors and flexors during motions in the three anatomical planes in healthy adults. The aim of this study was to explore characteristics of sEMG activities of these cervical muscles in nonspecific neck pain (NSNP) patients based on healthy adults. *Materials and Methods:* Participants were 24 NSNP patients (NSNP group, mean ± SD of age, 47.5 ± 15.5) and 24 healthy adults (control group, 20.5 ± 1.4). For each participant, sEMG of the cervical extensors and flexors was recorded during neck flexion, extension, bilateral lateral flexion, bilateral rotation, and at the neutral position in Phase I (the neck from the neutral position to the maximum range of motion), Phase II (at the maximum range of motion), and Phase III (from the maximum range of motion to the neutral position), yielding a total of 42 phases. A percentage of maximum voluntary contraction to normalize muscle activity in each phase was calculated to obtain the ratio of muscle activities in the extensors and flexors in each of 36 phases of the motions to the neutral position and ratio of the flexors to extensors in activity for 21 phases. *Results:* In 28 of 36 phases of the motions, the ratios of muscle activities in the extensors and flexors to the neutral position in the NSNP group were significantly larger than the control group (*p* < 0.05). In 6 of 21 phases, the ratios of the flexors to extensors in activity in the NSNP group were significantly larger than in the control group (*p* < 0.05). *Conclusions:* In NSNP patients, the activity of the cervical extensors and flexors associated with neck motion increased with an imbalance in activity between these muscles compared to their activity in healthy adults. The results of this study will be useful in understanding the pathogenesis of NSNP and in constructing an objective evaluation of the treatment efficacy on NSNP patients.

## 1. Introduction

Pain in the back and side of the neck between the superior nuchal line and the first thoracic spinous process vertebra without specific lesions and neurological symptoms is referred to as nonspecific neck pain (NSNP) [1], and this neck pain lasting more than 3 months is defined as chronic NSNP [2]. The term “nonspecific” is in contrast to “specific” neck pain caused by trauma or medical illness because the pathologic cause is unknown [3]. The lifetime prevalence of NSNP is estimated to be about 65% of the population [4], making it a global health issue. Although the pathogenesis of NSNP is still unrevealed, overstrain of the cervical muscles, which play an important role in maintaining neck stability, could be the main cause [5] since the cervical muscles are responsible for about 80% of the functional stability of the neck [6], and one of the initial symptoms of neck pain is stiffness of the cervical muscles [7,8].

For this reason, studies have been conducted using surface electromyography (sEMG) to measure the activity of the cervical muscles in order to explore the pathogenesis of NSNP. In fact, in a comparison of sEMG of the cervical muscles in NSNP patients and healthy adults during isometric contraction in the neutral head/neck position, the activity of the cervical muscles in NSNP patients was significantly larger than that in healthy adults, suggesting a relationship between hypertonia of the cervical muscles and NSNP [9]. In a study using sEMG to record the activity of the cervical extensors during the neck flexion from the neutral head/neck position to the maximum range of motion, holding the neck in that position, and returning the neck to the neutral head/neck position, a significantly increased activity of the cervical extensors while holding the maximally flexed position was found in NSNP patients compared to healthy controls [10,11,12]. In addition to the cervical extensors, increased activity of the cervical flexors during the craniocervical flexion test (CCFT) has been reported in NSNP patients [13,14].

On the other hand, it was revealed that the strength [15,16] and endurance [17,18] of the cervical muscles, which play an important role in neck stability, decreased in NSNP patients compared to healthy adults, and the physiological cross-sectional area of the cervical muscles is reduced [19]. Thus, while NSNP patients have increased activity in the cervical muscles, they have decreased strength and endurance in the cervical muscles, which suggests complexity of the pathophysiology of NSNP patients in terms of the cervical muscle activity.

To date, examination of the activity of the cervical muscles during flexion of the neck has been considered important for the evaluation of patients with neck pain in studies using sEMG [20]. However, the motion of the neck has a wide range of mobility, as there are flexion and extension in the sagittal plane, lateral flexion in the frontal plane, and rotation in the horizontal plane [21]. A motion in each direction or combinations of such motions in other directions are required during various functional tasks in daily life [22]. In particular, neck rotation is a risk factor for neck pain [23] and for sickness absence at work [24], which is noteworthy for considering the pathogenesis of NSNP. Accordingly, it is indispensable to understand the activity of the cervical muscles in the neutral position and motions in the sagittal, frontal, and horizontal planes of the neck in healthy adults without neck pain [25] in order to understand the complex behavior of the cervical muscle activity in NSNP and its relationship to pain. However, there had been no reports that investigated the comprehensive activity patterns of the cervical extensors and flexors in the neutral position and motions in healthy adults without neck pain, let alone NSNP patients [25].

Therefore, we investigated sEMGs in the neutral position and basic motions of the cervical extensors and flexors in the three anatomical planes in healthy adults without neck pain as a basis for understanding the pathophysiology of NSNP and for the development of criteria to evaluate treatment efficacy for neck pain [25]. In healthy adults, the extensors were significantly more active than the flexors in the neutral posture, flexion, extension, lateral flexion, and rotation to the same side the muscles were located [25]. The largest activity of the extensors and flexors from the neutral position to the maximum range of motion in each direction was during rotation. The behaviors of isometric and isotonic contraction in the neutral position and basic motions of the neck without artificial resistance reflected the daily motions of the neck under the gravity. Therefore, the behaviors of the cervical extensors and flexors obtained in our previous study are important basic data for understanding the pathophysiology of NSNP [25]. These basic data can be easily introduced into clinical practice because the measurement of sEMG of the cervical muscles with no resistance applied to the head is simpler than the measurement of sEMG of isometric contraction induced by applying resistance to the opposite direction of a motion at the neutral position of the head with a special device or manual force [26]. In other words, these data [25] make it possible to capture the specific sEMGs of the extensors and flexors during neck motions in NSNP patients.

Based on the above-mentioned scientific background, i.e., the pathogenesis of NSNP is unrevealed, NSNP is a global health issue, 80% of the functional stability of the neck is related to the cervical muscles, hypertonia of the cervical muscles but strength and endurance in these muscles decreased in NSNP patients, and the comprehensive activity patterns of the cervical extensors and flexors in healthy adults reflected daily motions [25], we examined activities of the cervical extensors and flexors in NSNP patients. In this study, sEMGs during the neutral position, flexion, extension, lateral flexion, and rotation of the neck were observed following the previous study to compare them to these muscle activity in healthy adults [25]. In addition, we observed the craniovertebral angle (CVA) in NSNP patients to examine whether CVAs were associated with activities of the cervical extensors in the neutral position by using the flexor–extensor ratio to explore the pathogenesis of NSNP from the perspective of poor posture.

The purpose of this study was to investigate characteristics of activities of the cervical extensors and flexors in NSNP patients from (1) the ratio of muscle activity in motions to muscle activity in the neutral position in the extensors and flexors and (2) the ratio of activity of the flexors to activity of the extensors in each motion of the neck in the sagittal, frontal, and horizontal planes. The null hypothesis of this study was that there was no difference in activity in the cervical extensors and flexors between healthy adults and NSNP patients. This study revealed the activity of the cervical extensors and flexors in NSNP patients, which could enable understanding the pathophysiology of NSNP and constructing an objective measurement of the effectiveness of treatment for neck pain.

## 2. Methods

This was a cross-sectional study in which the cervical muscle activity patterns with neck motions in healthy adults and NSNP patients was observed. The procedures for the measurement and analysis of cervical muscle activity and assessment of head and neck posture were performed according to the previous study [25]. This study was approved by the Ethics Committee of Tokyo Ariake University Medical and Health Sciences (approval no. 277, date of approval: 21 January 2019). Data from healthy adults and NSNP patients were recorded in the laboratory and the acupuncture clinic at Tokyo Ariake University Medical and Health Sciences, respectively. Participant recruitment began in January 2019, and all the data were obtained between January 2019 and April 2022.

### 2.1. Participants

#### 2.1.1. Participants of NSNP and Control Groups

We recruited participants for each group by posting flyers inviting them to participate in the study at the Acupuncture Clinic at Tokyo Ariake University of Medical and Health Sciences and at the university campus. Twenty-four patients complaining of NSNP (NSNP group) who came to the acupuncture clinic at Tokyo Ariake University Medical and Health Sciences and twenty-four healthy adults without pain in the neck and shoulder region (control group) participated in this study. Prior to participation in the study, it was checked by a research assistant whether each participant met the eligibility criteria for participation in the study. Subsequently, those who met the eligibility criteria for study participation were given a written and oral explanation of the content and purpose of the study, and their written consents were obtained prior to the start of the study. The inclusion and exclusion criteria for the study participants in the NSNP and control groups were as described below.

#### 2.1.2. Inclusion and Exclusion Criteria of the NSNP Group

Inclusion criteria: Patients with pain in the neck and/or shoulder.Exclusion criteria: Patients who had neither pain nor stiffness in the neck and shoulder; patients with neurological symptoms such as numbness or hypesthesia in the neck, shoulder, and upper extremity; and patients who suffered from cervical spine diseases such as cervical disc herniation, central and peripheral neuropathy, rheumatoid arthritis, or medical diseases [27].

#### 2.1.3. Inclusion and Exclusion Criteria of the Control Group

Inclusion criteria: Patients with neither pain nor stiffness in the neck, shoulder and/or upper extremity, and no neurological symptoms such as numbness or hypesthesia.Exclusion criteria: Patients who had pain or stiffness in the neck, shoulder, and upper extremity, or who had neurological symptoms such as numbness or hypesthesia, cervical spine disease such as cervical disc herniation, central and peripheral neuropathy, rheumatoid arthritis, or medical disease [25].

### 2.2. Procedures for Measuring Cervical Muscle Activity

#### 2.2.1. Surface Electromyogram (sEMG)

The activities of the bilateral cervical extensors and flexors were recorded using a sEMG (Neuropack X1: MEB-2306, NIHON KOHDEN CORPORATION, Tokyo, Japan) [25]. The skin where the electrodes were to be placed was disinfected with alcohol-soaked cotton after treatment with a skin pretreatment agent to decrease the contact resistance between the electrode and the skin. A pair of disposable Ag/AgCl surface electrodes (NSC electrode, NM-317Y3, NIHON KOHDEN CORPORATION, Tokyo, Japan), which were positioned 2 cm apart, was then placed bilaterally on the cervical erector spinae muscle approximately 2 cm lateral to the fourth cervical spinous process for recording of the cervical extensor activity [12,25,28]; the pair of electrodes was placed one-third of the way from the suprasternal notch to the mastoid process on the line connecting the mastoid process to the suprasternal notch for the recording of the cervical flexor activity [25,29]. Ground electrodes were put on the clavicle and scapular spine. One research assistant applied the electrodes to all participants to minimize bias due to differences in the placement of electrodes among patients. The muscle activity was measured under a sampling frequency of 1000 Hz and a bandpass filter of 20–500 Hz [13,30]. Recorded muscle activity was imported into data analysis software (LabChart Pro8, ADInstruments Japan, Nagoya, Japan), and full-wave rectification was performed [25].

#### 2.2.2. Procedures for Measuring Muscle Activity

For the recording of sEMG, participants sat in a chair with their back straight, the knees and hips flexed to 90°, and soles of the feet on the floor. The head and neck were set at 0°, the eyes were set to look straight ahead, the chin was pulled back, and the upper limbs were relaxed [12,25,31,32]. This posture was defined as the “neutral position” [25]. Following our previous study, the muscle activity of the cervical extensors and flexors was measured in the following order: holding the neutral position, flexion and extension in the sagittal plane, right and left lateral flexion in the frontal plane, and right and left rotation in the horizontal plane, each of which was performed over 9 s. Each 9 s period of the above motions consisted of three phases of 3 s as follows: Phase I, moving the head from the neutral position to the maximum range of motion; Phase II, holding the head at the maximum range of motion; Phase III, returning the head from the maximum range of motion to the neutral position [10,11,25,31]. The participant performed each motion consisting of the three phases according to the research assistant’s count at a rate of one time per second.

After all the above measurements were performed, the maximal isometric voluntary contractions (MVCs) in each direction for all the motions at the neutral position were measured for 5 s against manual resistance by the research assistant to normalize the amplitude of the sEMG during each motion [25,26]. The sEMG for the neutral position, each motion, and MVC were measured once for the NSNP group considering the burden on the patient and three times each for the control group. To prevent the influence of muscle fatigue, a 2 min rest was provided between each measurement [25].

### 2.3. Assessment of Head and Neck Posture

The head and neck posture of seated participants was evaluated on images in the sagittal plane taken with a digital camera (D5300, NIKON CORPORATION, Tokyo, Japan) positioned at 1.5 m from the participant’s left side adjusted at shoulder height [25,33,34]. To evaluate the posture by CVA, the research assistant affixed markers to the two reference points of the participant’s left tragus of the ear and the seventh cervical spinous process (C7) before taking pictures [25]. The research assistant then took three images of the participants in the neutral position.

### 2.4. Evaluation of NSNP

Each patient was asked to rate the neck pain intensity on the numerical rating scale (NRS: 0 being no pain and 10 being the most severe pain imaginable) and to record the Japanese version of the neck disability index (NDI), which is a self-administered questionnaire for neck pain.

### 2.5. Data Analysis

#### 2.5.1. Calculation of Grand Ensemble Average

For data in individual healthy participants, ensemble averages of sEMGs of the cervical extensors and flexors in the neutral position and each motion were generated by three rectified sEMGs that were converted from the three sEMGs [25]. For a grand ensemble average of all healthy participants, the ensemble averages of individual participants were summed and averaged for the same muscles for the neutral position and each motion [25,35,36,37]. For data in NSNP patients, i.e., the grand ensemble average of all NSNP patients, the rectified sEMGs of individual NSNP patients were summed and averaged for the same muscles for the neutral position and each motion. The grand ensemble averages in healthy participants and NSNP patients were depicted for the extensors and flexors in the neutral position and each motion for visual comparison of both groups [25].

#### 2.5.2. Calculation of Muscle Activity Ratio

The integrated sEMG (iEMG) of each of the bilateral cervical extensors and flexors was calculated for each of the three phases from the sEMG during the neutral position and each motion in the control group [25]. For MVC, the iEMG of each of the bilateral cervical extensors and flexors was calculated from the sEMG for the middle 3 s of the 5 s of each motion [25,38,39,40]. In the NSNP group, the iEMG value was obtained from one measurement in each motion. In the control group, the mean of the three iEMGs for each motion was used. The iEMGs of the cervical extensors and flexors in each phase in the neutral position and each motion were then normalized to %MVC by dividing the largest iEMGs of MVCs among the left and right extensors and flexors during all motions [25].

Then, for flexion and extension, the left and right %MVCs were combined in the extensors and flexors, respectively, as the extensors and flexors group [25]. The %MVC of the right extensors during right lateral flexion and the left extensors during left lateral flexion were combined as the %MVC of the ipsilateral extensors of lateral flexion, and the %MVC of the left extensors during right lateral flexion and the right extensors during left lateral flexion were combined as the %MVC of the contralateral extensors of lateral flexion [25]. The same process was performed for the cervical flexors during lateral flexion and the cervical extensors and flexors during rotation [25]. For lateral flexion and rotation, therefore, the motions to the side of the cervical extensors and flexors located were referred to as ipsilateral lateral flexion and ipsilateral rotation, and motions to the side opposite to these muscles located were referred to as contralateral lateral flexion and contralateral rotation [25].

The ratio of muscle activity in each phase of each motion to the neutral position for the extensors and flexors was calculated by dividing the %MVC of each phase of each motion by the %MVC in the neutral position; the flexor–extensor ratio was obtained by dividing the %MVC of the flexors during each phase of the neutral position and each motion by the corresponding %MVC of the extensors [25]. 

#### 2.5.3. Calculation of CVA

On each image taken, the CVA, which is the angle formed by the horizontal line through C7 and the line connecting the midpoint of the tragus and C7, was calculated using ImageJ (NIH), which is one of the objective methods to evaluate the head and neck posture in the sagittal plane [25,33,34,41,42]. The CVA of each participant was a mean of the three CVAs calculated from the three images [42,43].

### 2.6. Statistical Analysis

A Kolmogorov–Smirnov normality test showed that ratios of muscle activity of the motions to the neutral position in 7/18 phases in the control and 1/18 in NSNP group for the extensors and 1/18 in the control and 1/18 in NSNP group for the flexors; and for flexor–extensor ratios, 4/21 phases in the control group were normally distributed, but the remaining phases were not normally distributed. Therefore, we performed nonparametric analysis and supplemented it with parametric analysis. Statistical analysis was performed using SPSS version 28 (IBM Japan, Ltd., Tokyo, Japan). Mann–Whitney U test was performed for comparisons of the ratio of muscle activity in each phase of the motion to the neutral position, the flexor–extensor ratio and CVA between the NSNP and control groups. Spearman’s rank correlation coefficient was used to determine a correlation between the flexion–extension ratio in the neutral position and CVA. The parametric analysis by independent *t*-test, which had been used in a previous study, was performed as supplementary analysis [25]. Statistical significance was set at *p* < 0.05. The sample size was not calculated since there have been no sEMG studies using the same index as in the present study comparing NSNP patients with healthy adults to estimate an effect size. Therefore, we calculated a power in the comparisons of the ratios of muscle activities in each phase of each motion to the neutral position and comparison of flexor–extensor ratios between the NSNP and control groups for each phase.

## 3. Results

### 3.1. Demographics

Twenty-seven patients with neck pain agreed to participate. Except for 3 patients who were excluded because they matched the exclusion criteria (rheumatoid arthritis, *n* = 1; hypertension, *n* = 2), 24 patients (16 male, 8 female) were included in the NSNP group. NDI score and NRS were 10.8 ± 4.7 and 5.2 ± 1.7, respectively, in the NSNP group. All NSNP patients felt stiffness to some extent in the neck. Twenty-four participants (7 male, 17 female) for the control group were included. Demographic characteristic data of participants in the NSNP and control groups are shown in Table 1. The age in the NSNP group was significantly higher than the control group (*p* < 0.001). The height (*p* = 0.040), weight (*p* = 0.005), and BMI (*p* = 0.020) in the NSNP group were significantly less than control group. 

### 3.2. Grand Ensemble Average in Healthy Participants and NSNP Patients

The grand ensemble averages of the cervical extensors and flexors in the neutral position and each motion in the control and NSNP groups are shown in Figure 1. The cervical extensors and flexors activities in NSNP patients were larger than in healthy participants, especially the activity of the flexors during flexion, extension, and ipsilateral rotation.

### 3.3. Ratio of Muscle Activity of Each Motion to the Neutral Position in the NSNP and Control Groups

Statistical differences in the ratio of muscle activity in the motions to the neutral position in each phase between the NSNP and control groups for the cervical extensors and flexors are shown in Figure 2.

For the extensors, ratios in the NSNP were significantly larger than the control group in flexion (Phase III, *p* = 0.018); in extension (Phase I, *p* < 0.001; Phase II, *p* = 0.025; Phase III, *p* < 0.001); in ipsilateral lateral flexion (Phase I, *p* < 0.001; Phase II, *p* < 0.001; Phase III, *p* = 0.001); in contralateral lateral flexion (Phase I, *p* < 0.001; Phase II, *p* = 0.004; Phase III, *p* = 0.030); in ipsilateral rotation (Phase I, *p* = 0.042); and in contralateral rotation (Phase I, *p* < 0.001; Phase III, *p* = 0.018).

For the flexors, ratios in the NSNP were significantly larger than the control group in flexion (Phase I, *p* < 0.001; Phase II, *p* = 0.035; Phase III, *p* = 0.002); in extension (Phase I, *p* = 0.001; Phase II, *p* < 0.001; Phase III, *p* = 0.009); in ipsilateral lateral flexion (Phase I, *p* < 0.001; Phase III, *p* = 0.024); in contralateral lateral flexion (Phase I, *p* = 0.001; Phase II, *p* = 0.002; Phase III, *p* < 0.001); in ipsilateral rotation (Phase I, *p* < 0.001; Phase II, *p* < 0.001; Phase III, *p* < 0.001); and in contralateral rotation (Phase I, *p* = 0.024).

### 3.4. Flexor–Extensor Ratios in the NSNP and Control Groups

Statistical differences in the ratio of flexor activity to extensor activity (flexor–extensor ratio) were revealed between the NSNP and control groups for each phase of each motion, as shown in Figure 3.

The ratios in the NSNP group for flexion in Phase I (*p* = 0.020), extension in Phase II (*p* = 0.039), contralateral lateral flexion in Phase III (*p* = 0.005), and ipsilateral rotation in all phases (*p* = 0.021 for Phase I, *p* < 0.001 for Phases II, and *p* = 0.002 for Phase III) were larger than the control group.

### 3.5. CVA in the NSNP and Control Groups

#### 3.5.1. Comparison of the NSNP and Control Groups

No statistically significant difference in CVA between the two groups was found (*p* = 0.538).

#### 3.5.2. Correlation of CVA and Flexion–Extension Ratio in the Neutral Position

There was a positive correlation found between CVA and flexion–extension ratio of the middle 3 s (Phase II) in the neural position (*ρ* = 0.393, *p* = 0.006) in NSNP patients (Figure 4).

## 4. Discussion

In this study, we measured sEMGs of the cervical extensors and flexors of healthy adults (control group) and patients with NSNP (NSNP group) in the neutral position and during flexion, extension, lateral flexion, and rotation of the neck (three phases in the neutral position and three phases in each of the six motions in the extensors and flexors, yielding a total of 48 phases). The ratio of muscle activity in the motions to the neutral position in each phase and the flexor–extensor ratio in each phase was determined. In 28 of the 36 phases, including the extensors and flexors, the ratio of muscle activity in the motions to the neutral position and in 6 of the 21 phases for the flexor–extensor ratio, including the neutral position in the NSNP group, were significantly larger than the control group. 

### 4.1. Ratio of Muscle Activity in Motions to the Neutral Position in Each Phase in the Extensors and Flexors

In the NSNP group in the present study, the ratio of the muscle activity in motion to the neutral position in the cervical extensors and flexors was significantly increased in approximately 78% of all phases compared to the control group. These results suggest that the activity of the cervical extensors and flexors increased during neck motion in NSNP patients. The results indicate that the evaluation method using sEMG during neck motions in our previous [25] and present studies is useful for understanding the pathophysiology of NSNP from the aspect of muscle activity. It was reported that NSNP patients showed increased activity of the extensors during flexion [11,12,20] and the superficial flexors during CCFT compared to healthy adults in previous studies using sEMG [13,14]. Moreover, increases in both extensor and flexor activities during CCFT in NSNP patients were reported [44]. The current results supporting these previous findings suggest that the sEMG of the cervical extensors and flexors in the neutral position and during flexion, extension, lateral flexion, and rotation of the neck can be a useful tool for physician, physiotherapists, acupuncturists, manual therapists, etc., to evaluate NSNP patients and the effectiveness of their treatment of NSNP patients with objectivity and relative ease in the clinical field.

In our previous study employing healthy adults, we compared muscle activities in the 36 phases in the motions of the extensors and flexors to activities in the neutral position measured using the same procedure as in this study [25]. We found 23 phases that showed a significant increase comparing with the neutral position but 13 phases that did not [25]. In 16 of these 23 phases, the ratios of muscle activity to the neutral position in the NSNP group was significantly larger than in the control group in this study. Furthermore, in 12 of the 13 phases], the NSNP group showed significantly larger ratios of muscle activity to the neutral position compared with the control group. These results indicate a general increase in activity of the cervical extensors and flexors in NSNP patients. This general increase in cervical muscle activity can be considered to reflect a modulation of the neck kinematic impairment, which occurred as a defensive response to neck pain associated with motion, to carry out the motion while maintaining its stability and inhibiting neck motion. The modulation in motor control is believed to continually give rise to abnormal loads on the tissues to cause mechanically provoked pain [45,46,47]. Since there is little contribution of ligaments to neck stability, the cervical muscles are thought to play an important role [6], especially the synergy of the superficial and deep cervical muscle activity, for the stability of the neck [48]. In NSNP patients, the function of the deep cervical muscles may decrease because the activity of the deep cervical muscles plays a central role in neck stability [49], which decreased during CCFT compared to healthy adults [50], and the physiological cross-sectional area of the deep cervical extensors diminished [19]. It was also reported that the activity of the superficial flexors increased as a compensatory operation against the decrease in stability due to the decrease in the activity of the deep flexors in NSNP patients [50,51]. The increased sEMG of the superficial cervical muscles observed in the NSNP patients in the present study might compensate for the decreased activity of the deep muscles to prevent the associated loss of stability. The 13 phases that did not show a significant increase in activity comparing with the neutral position in the previous study [25] suggest that the activity in these phases arose from the antagonist muscles. If so, almost all antagonist muscle activity increased in NSNP patients. Spine stability is thought to be enhanced by increasing activity of the antagonist muscle as well as the agonist muscle located around the spine [52,53,54]. In addition, chronic NSNP patients were reported to have increased activity of the antagonist muscle during isometric contraction at the neutral head/neck position [55]. In the aspect of the agonist muscle, which produces a moment in the same direction of joint movement, and the antagonist muscle, which generates a moment in the direction opposite to joint movement [56,57], the increased activity of the antagonist muscles observed in NSNP patients in the present study is considered to enhance cervical stability. 

However, the increased activity of muscles to stabilize the neck as a protective response to pain has unfavorable effects on cervical motions by an increase in spinal compression [58], excessive compressive load on the cervical facet joint [19], and muscle fatigue [59]. Chronic low-back pain was reported to be induced by prolonged compensation with muscle activity to maintain the mechanical stability of the spine [60,61]. In NSNP patients, the compensatory motor control, especially for the increasing antagonist muscle activity, to bring about these unfavorable effects may create abnormal tissue loading to cause or exacerbate the mechanically provoked pain.

### 4.2. Flexor–Extensor Ratio in Each Phase in Each Motion

In 6 of the 21 phases, the flexor–extensor ratio in the NSNP group was significantly larger than the control group. For these six phases, the activity of the extensors was larger than the flexors in the healthy adults [25]. The flexor–extensor ratio in healthy adults was not maintained during these phases of neck motion in the NSNP group. In other words, there was a larger increase in flexor activity relative to the increase in extensor activity in NSNP patients. This suggests that the mode of motions of the cervical muscles was altered to give rise to an imbalance between the extensors and flexors in the NSNP patients. Similar to imbalance of muscle strength between the flexors and extensors in the trunk acting as a risk factor for low back pain [62], imbalance of muscle activity between cervical flexors and extensors during isometric contraction seen in migraine patients [30], and imbalance of activity in the cervical muscles disrupting the mechanical equilibrium of the spine to increase in spinal shear forces [63], imbalance between the cervical extensors and flexors may result in excessive mechanical stress on the neck to cause or exacerbate pain in NSNP patients.

### 4.3. Relationship between Neck Posture and Cervical Muscle Activity

In NSNP patients, a positive correlation was found between CVA, which is an objective measure of head and neck posture in the sagittal plane [25], and the flexor–extensor ratio in the neutral position. That is, a larger increase in extensor activity relative to flexor activity resulted in a smaller CVA. Since the head and cervical spine are located in front of the center of gravity line, the activity of the extensors located on the posterior of the neck plays a major role in maintaining the head and cervical spine in the neutral position against the external torque due to gravity, subjecting the head to forward flexion. In fact, increased activity of the extensors with the increase in external torque due to the decrease in CVA was revealed in healthy adults [25]. The same relationship between CVA and the activity in the extensors in healthy people was observed in NSNP. Instructing NSNP patients to maintain a posture that increases CVA to reduce overload on the extensors in daily life may be an effective means of relieving neck pain.

### 4.4. Clinical Significance

The activity of the cervical muscles during neck flexion was used as an indicator of NSNP patients in many studies [11,12,20] because it is considered important for the assessment of NSNP patients [20]. However, the ratios of activities of motions to the neutral position in the extensors and flexors and the flexor–extensor ratios in the NSNP group were significantly different from the control group in the motions on the three anatomical planes (sagittal, frontal, and horizontal planes). These results suggest that muscle activity not only in neck flexion but in all neck motions on the three anatomical planes could be useful in evaluating NSNP patients. Among them, an assessment of rotation that is considered a risk factor for neck pain [23,24] is more important because the largest activities in the extensors and flexors were observed among the six motions in the previous study [25], and significant differences in flexor–extensor ratios were found in all three phases in ipsilateral rotation between the NSNP and control group. The results of this study revealed two characteristic muscle activities in NSNP patients: a general increase in activity of the cervical extensors and flexors and the imbalance in the activity of the extensors and flexors. Therefore, a decrease in the activity of the cervical extensors and flexors and improvement in the imbalance between the extensors and flexors could be an indication for the evaluation of treatment of NSNP patients. 

### 4.5. Limitations and Strengths

One of the limitations of this study is that the ages of the participants between the control and NSNP groups were not matched. The mean age of the NSNP group in this study was 47.5 years, consisting of middle-aged and older participants, which is the age group with the most people complaining of neck pain [64]. However, recruiting age-matched middle-aged healthy adults as the control group was difficult due to resource limitations. Moreover, there was a significant difference in body weight, height, and BMI between the control and NSNP groups, and the participants in the NSNP group were only patients who sought acupuncture treatment; therefore, caution is needed for generalizing the current results to all NSNP patients, including NSNP patients of different ages, body weight, height, races, countries, and who seek other treatments. Furthermore, it is not possible to determine whether the ratio of muscle activity of motions to the neutral position and the flexor–extensor ratio changes observed in the NSNP group were a result of NSNP or a cause of NSNP only through this study. Lastly, sEMG measurement was once for each phase in NSNP patients considering the patient’s burden. The present results should be interpreted with these limitations in mind, and a longitudinal study should be conducted to address these issues in the future. On the other hand, we believe a strength of this study was the measurements of sEMG of isotonic cervical muscle activities in the neck motions reflecting the motions in daily life, which cannot be obtained from sEMG studies observing isometric contractions with an artificial resistance.

### 4.6. Overall Interpretation of the Results

To understand the pathophysiology of NSNP and develop an objective evaluation of the treatment efficacy on NSNP, we first studied sEMG activities of the cervical extensors and flexors in the neutral position and during the basic motions in the frontal, sagittal, and horizontal planes in healthy adults [25]. Given the present results in conjunction with the previous study [25], increased activity of the cervical extensors and flexors in NSNP patients comparing to health adults reflected muscle behaviors in NSNP patients.

In our studies, we divided the cervical muscles into the extensors and flexors to study cervical muscles activities. The results did not reflect the details of activities in the individual cervical muscles. Although it is ideal to study the activities in individual cervical muscles as many as possible, it would be difficult to implement such a time-consuming method into clinical practice. Therefore, representative cervical muscles have been usually investigated in previous sEMG studies [9,10,11,12,13,14,30,31,32,48,49,50,51]. Considering the relatively simple, less time-consuming, and less burdensome nature of the current method focusing on the two muscle groups, the methods using sEMG in this study have a certain rationality for their introduction into clinical field.

The treatment efficacy on NSNP patients was assessed by subjective indices, such as a visual analogue scale or a numerical rating scale, etc. [65,66,67,68,69], which could be influenced by biases attributed to patients, treatment providers, and evaluators [70,71]. Introducing sEMG as objective indication provides more accurate evaluation of NSNP patients. By comparing sEMG activities and the subjective indices, it is possible to evaluate the treatment efficacy on NSNP patients with more objectivity. In other words, it is now possible to evaluate NSNP patients by finding increased activities in the cervical extensors and flexors and examine the relationship between neck pain and the increased muscle activities.

## 5. Conclusions

In NSNP patients, the activity of the cervical extensors and flexors associated with neck motion in the frontal, sagittal, and horizontal planes was increased, with an imbalance of activity between the extensors and flexors compared to their activity in healthy adults.

## Figures and Tables

**Figure 1 medicina-58-01770-f001:**
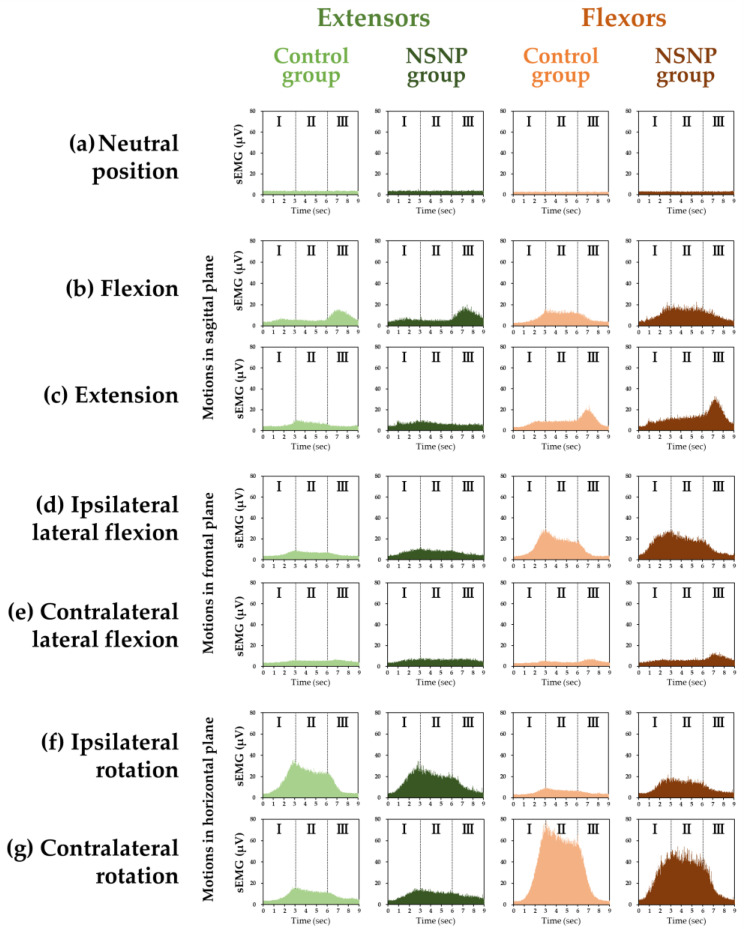
The grand ensemble average of sEMGs (surface electromyograms) of the cervical extensors and flexors (**a**) in the neutral position and during neck (**b**) flexion, (**c**) extension, (**d**) ipsilateral lateral flexion, (**e**) contralateral lateral flexion, (**f**) ipsilateral rotation, and (**g**) contralateral rotation in the control group and NSNP (nonspecific neck pain) group. “I” indicates “Phase I”, a 3 s period during a motion from the neutral position to the maximum range of motion. “II” indicates “Phase II”, a 3 s period of maintaining the neck at the maximum range of motion. “III” indicates “Phase III”, a 3 s period during the motion from the maximum range of motion to the neutral position. (**b**,**c**) Motion in the sagittal plane; (**d**,**e**) motion in the frontal plane; (**f**,**g**) motion in the horizontal plane. In each motion, the greenish color indicates the activity of the extensors, and the brownish color indicates the activity of the flexors. Dark green and brown indicate activity in the NSNP group; light green and brown indicate activity in the control group.

**Figure 2 medicina-58-01770-f002:**
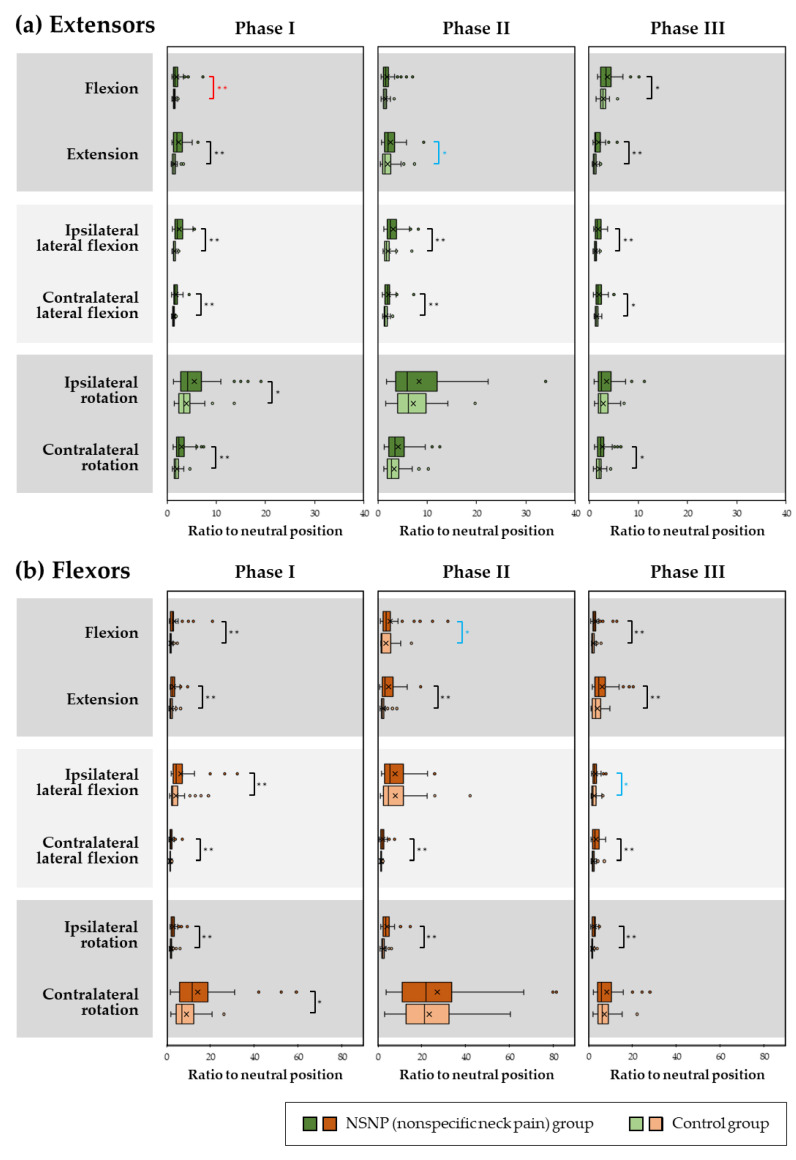
Ratio of muscle activity in the motions to the neutral position in each phase for the cervical (**a**) extensors and (**b**) flexors in the NSNP and control groups. For the extensors, the ratios in all phases in flexion, Phase I in extension, Phase I in contralateral lateral flexion, Phase II in ipsilateral rotation and Phase III in contralateral rotation in the control group, and Phase III in flexion in the NSNP group were normally distributed but not for the remaining phases. For the flexors, the ratios in Phase II in contralateral rotation in the control group, and Phase III in ipsilateral rotation in the NSNP group were normally distributed but not for the remaining phases. Red asterisks indicate a significant difference in only parametric analysis, blue asterisks indicate a significant difference in only nonparametric analysis, and black asterisks indicate a significant difference in both analyses. The top, middle, and bottom lines of the boxes correspond to the 75th, median, and 25th percentile, respectively. The whiskers extend from the minimum to maximum. The cross marks indicate the arithmetic means. The powers were less than 0.8 in 15 phases (extensors: flexion, extension, and contralateral rotation in Phase II; ipsilateral rotation in phase I, II, and III; flexors: flexion, extension, and contralateral rotation in Phase II and III; ipsilateral lateral flexion in phase I, II, and III). The rest of the 21 phases were *β* ≥ 0.8. * indicates, *p* < 0.05 and ** indicates *p* < 0.01.

**Figure 3 medicina-58-01770-f003:**
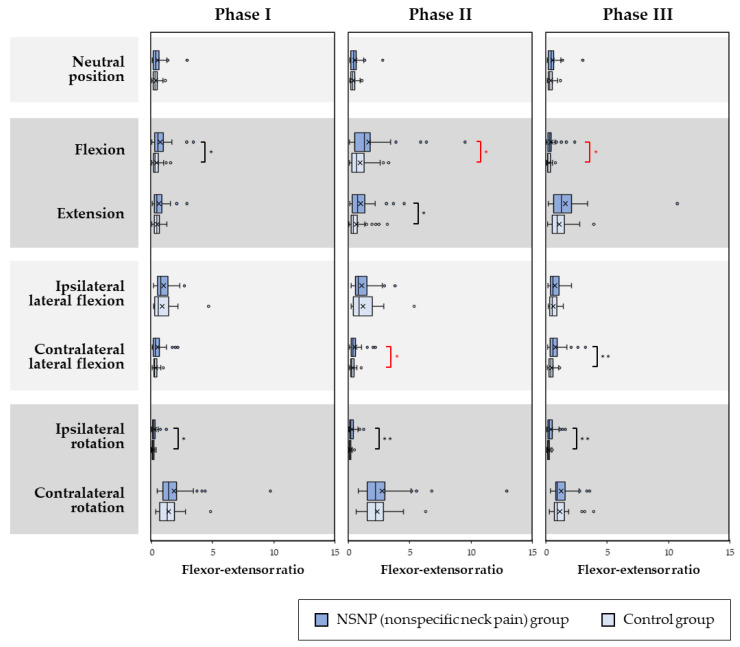
Flexor–extensor ratio in each phase in the NSNP and control groups. The ratios in Phase I in extension, Phase I in ipsilateral rotation. and Phase I and II in contralateral rotation in the control group were normally distributed but not for the remaining phases. Red asterisks indicate a significant difference in only parametric analysis, and black asterisks indicate a significant difference in both analyses. The top, middle, and bottom lines of the boxes correspond to the 75th, median, and 25th percentile, respectively. The whiskers extend from the minimum to the maximum. The cross marks indicate the arithmetic means. The powers were less than 0.8 in 15 phases (neutral position, extension, ipsilateral lateral flexion, and contralateral rotation in phase I, II, and III; flexion in phase II and III; contralateral lateral flexion in phase I). The rest of the six phases were *β* ≥ 0.8. * indicates *p* < 0.05 and ** indicates *p* < 0.01.

**Figure 4 medicina-58-01770-f004:**
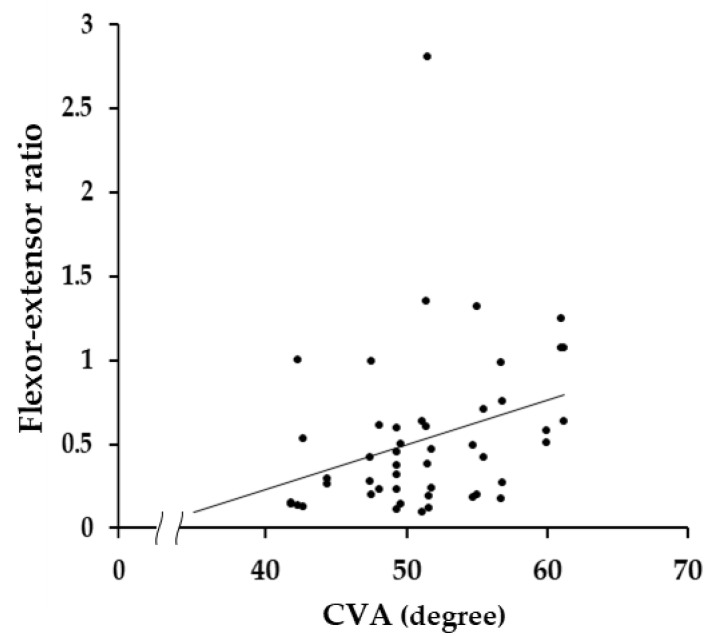
The correlation between craniovertebral angle (CVA) and flexor–extensor ratio of Phase II in the neutral position in the NSNP (nonspecific neck pain) group. A larger flexor–extensor ratio indicates a relatively lower extensor activity. There was a significant positive correlation (*ρ* = 0.393, *p* = 0.006).

**Table 1 medicina-58-01770-t001:** Demographic characteristic data of participants.

		NSNP ^1^ Group (*n* = 24)	Control Group (*n* = 24)	*p*-Value
Age (years)	mean ± SD	47.5 ± 15.5	20.5 ± 1.4	<0.001
	95% CI	21.2–43.8	20.2–20.8	
Height (cm)	mean ± SD	160.8 ± 7.9	165.5 ± 8.1	0.040
	95% CI	158.9–162.7	163.6–167.4	
Weight (kg)	mean ± SD	54.4 ± 10.4	61.9 ± 9.5	0.005
	95% CI	52.0–56.9	59.7–64.2	
BMI ^2^ (kg/m^2^)	mean ± SD	20.9 ± 2.8	22.5 ± 2.6	0.020
	95% CI	20.3–21.6	21.9–23.1	

^1^ NSNP, nonspecific neck pain; ^2^ BMI, body mass index.

## Data Availability

The data presented in this study are available on request from the corresponding author.

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
