# Peer review of "Characteristics of Surface Electromyograph Activity of Cervical Extensors and Flexors in Nonspecific Neck Pain Patients: A Cross-Sectional Study"

_medicina, 2022, doi:10.3390/medicina58121770_

Round 1

Reviewer 1 Report

Medicina-2009423

Review: Characteristics of Activity of Cervical Extensors and Flexors in Nonspecific Neck Pain Patients: A Cross-Sectional Study

Proposed: major revision

Dear editor,

Thank you for letting me take part of, and review this manuscript. The topic is interesting and relevant. The manuscript also suits well for the scope of the journal and most parts are well written. Further, it contributes to knowledge within the area of nonspecific neck pain. However, there are shortcomings in this manuscript:

Introduction:

From the aim it is not clear why later on correlations has been performed in this study.

Methods:

In this study a NSNP group was compared with a control group, however, it is not clear to me why the data are also compared with data of former research with healthy persons.

From my understanding the control group of this study consists of healthy persons, so I don’t understand why an extra healthy group of a former study was added.

A power calculation is missing.

Please check/test if the found data were normally distributed. When your data are (partly) not normally distributed, you have to test these data with non-parametric statistical tests. Then also the median and interquartile ranges have to be shown.

Results:

The results has to be rewritten. It seems as if someone else has written this part of the study.

Please provide readable text and not a listing of found results.

There is a significant difference in age between the NSNP group and the control group; 1) add a Table to your results with the demographics and compare these groups statistically.

From Figure 1 the outcomes of the healthy participants are not clear, because the refereeing colours are not visible.

Insight in the exact results is missing. So please provide the outcomes of Figure 2 and 3 in a Table(s) with the means (/medians) standard deviations (/inter quartile ranges), 95% confidence intervals, and found p-values for the ANOVA and Bonferroni tests.

In part 3.3.2. p-values are given, however, it is not clear from the text which group performed significantly better.

As said related to the aim in the introduction, it is not crystal clear why these correlations are calculated.

Discussion:

The strengths of this study are missing in 4.5. It is better to describe both the strengths and limitations of your study.

Implications for future research is missing.

Also the implications for the clinic are missing. What can physicians, physiotherapists etc. learn from the found results in this study?

Author Response

Responses to Reviewer 1 comments
Thank you for taking time to peer review and for your valuable comments and suggestions. We
have made revisions according to each comment. We hope our revisions are satisfactory.

Here are our answers.

Introduction:

From the aim it is not clear why later on correlations has been performed in this study.

Response

According to the comment, we added the following sentence in the second last paragraph in the
Introduction.

Also, we observed the craniovertebral angle (CVA) in NSNP patients to examine whether the
correlation between CVAs and the flexorextensor ratio in the neutral position to explore the
pathogenesis of NSNP from the perspective of poor posture.

Methods:

In this study a NSNP group was compared with a control group, however, it is not clear to me
why the data are also compared with data of former research with healthy persons. From my
understanding the control group of this study consists of healthy persons, so I don’t understand
why an extra healthy group of a former study was added.

Response

According to the comment, we added the following text in the second last paragraph in the
Introduction.

Note that we conducted analysis using the data from healthy adults obtained in the previous study
[25] to ensure that the data from healthy adults in this study were an appropriate control, and to
avoid recruiting additional healthy adults in future studies for comparison with NSNP patients by
utilizing data of the reference group.

A power calculation is missing.

Response

Since there have been no previous sEMG studies comparing NSNP patients with healthy adults
using the same index as in the present study, a sample size calculation is not possible. Therefore, we
collected as many participants as possible within the recruitment period.

According to the comment, we added the following text in the 2.6. Statistical Analysis.

The sample size was not calculated since there has been no sEMG studies using the same index as in
the present study comparing NSNP patients with healthy adults to estimate an effect size.
Therefore, we calculated a power in the comparisons of the ra-tios of muscle activities of each
motion from the neutral position and comparison of flexor-extensor ratios among the NSNP, control
and reference groups for each phase.

According to the comment, we added the following text in the legend of Figure 2.

The powers were less than 0.8 in 5 phases (extensors: flexion, ipsilateral rotation, and contralateral
rotation in Phase II; flexors: ipsilateral flexion and contralateral rotation in Phase II) of 6 phases in
which there were no significant differences among the three groups. The rest 31 phases were β ≥ 0.8.

According to the comment, we added the following text in the legend of Figure 3.

The powers were less than 0.8 in 11 phases (neutral position and ipsilateral lateral flexion in phase I,
II and III, contralateral rotation in phase II and III, contralateral lateral flexion in phase I, ex-tension
in phase II, and flexion in phase III) of 14 phases in which there were no significant dif-ferences
among the three groups. The rest 10 phases were β ≥ 0.8.

Please check/test if the found data were normally distributed. When your data are (partly) not
normally distributed, you have to test these data with non-parametric statistical tests. Then also
the median and interquartile ranges have to be shown.

Response

Although normal distribution did not cut across all data, we performed the statistical analysis using
parametric tests in accordance with previous studies. However, we agree with the comments and
redone the statistical analysis using nonparametric. Also, we prepared the results and figures based
on the nonparametric analysis, in which significant differences found by parametric comparison
were inserted as supplementary. 2.6. Statistical analysis was also revised accordingly as follows.

Kruskal-Wallis test A one-way analysis of variance (ANOVA) was performed for the ratio of muscle
activity in each phase of the motion to the neutral position and the flexorextensor ratio and CVA in
the three groups, and a Dunn test Bonferroni test was performed for multiple comparisons when
significant differences were found. Spearman's rank correlation coefficient Pearson’s correlation
coefficient was used to determine a correlation between the flexionextension ratio in the neutral
position and CVA. The parametric analyses, a one-way analysis of variance (ANOVA) and a Bon-
ferroni test and Pearson’s correlation coefficient, which have been used in previous studies were
performed as supplementary analysis.

Results:

The results has to be rewritten. It seems as if someone else has written this part of the study.

Please provide readable text and not a listing of found results.

Response

According to
the comment, we revise the Results section.

Reviewer 2 Report

Reviewer Comments

Thank you very much for the opportunity to review the manuscript submission entitled: Characteristics of Activity of Cervical Extensors and Flexors in Nonspecific Neck Pain Patients: A Cross-Sectional Study.

The current study aims to explore characteristics of sEMG activities of these cervical muscles in nonspecific neck pain (NSNP) patients based on healthy adults.; The study is interesting; however, some limitations and constructive comments are pointed out below:

Specific comments

The text must be proofread in English to minor grammatical errors.

Title and abstract:

·      Title should include surface electromyogram activity rather than just activity.

·      Include p values for difference between groups.

·      End the abstract with clinical significance of the study

·      Include MeSH terms as keywords.

Introduction

·      Explain the scientific background and rationale for the investigation being reported.

·      State specific prespecified hypotheses

Methods

·      Present key elements of study design early in the paper

·      Clearly describe the diagnostic criteria for NSNP.

·      Describe the setting, locations, and relevant dates, including periods of recruitment and data collection.

·      Give the clear eligibility criteria, and the sources and methods of selection of participants.

·      Clearly Explain how the study size was arrived at.

·      Justify the use of statical tests used. Did the data follow normal distribution?

Discussion

·      Give a cautious overall interpretation of results considering objectives, limitations, multiplicity of analyses, results from similar studies, and other relevant evidence

·      Discuss the generalisability (external validity) of the study results

Author Response

Responses to Reviewer 2 comments
Thank you for taking time to peer review and for your valuable comments and suggestions. We
have made revisions according to each comment received. I would like to thank you again for your
kind consideration.

Here are our answers.

Specific comments:

The text must be proofread in English to minor grammatical errors.

Response

Thank you for the comment. Before submitting this paper, we have had it edited by MDPI's English
editing service. I will ask the editor how to handle this matter.

Title and abstract:

Title should include surface electromyogram activity rather than just activity.

Response

We revised the title according to the comment.

Include p values for difference between groups.

Response

P values has been added in accordance with the comment.

End the abstract with clinical significance of the study

Response

The following text has been added in accordance with the comment.

The results of this study will be useful in understanding the pathogenesis of NSNP and in
constructing an objective evaluation of the treatment efficacy on NSNP patients.

Include MeSH terms as keywords.

Response

Thank you for your comment, we have already included MeSH terms such as cervical vertebrae,
neck muscles, electromyography, etc., so we will leave the current keywords as they are.

Introduction:
Explain the scientific background and rationale for the investigation being reported.

Response

We have stated the scientific rationale for this study throughout the 1-5 paragraphs of the 1.
Introduction, however, we have added the following text to the last paragraph of the 1. Introduction
in accordance with comments.

Based on above-mentioned scientific background that the pathogenesis of NSNP is unrevealed,
NSNP is a global health issue, 80% of the functional stability of the neck is responsible to the
cervical muscles, the relationship between hypertonia of the cervical muscles and NSNP but
strength and endurance in the cervical muscles decreased were suggested, and the comprehensive
activity patterns of the cervical extensors and flexors in healthy adults reflected the daily motions
were revealed [25],

State specific prespecified hypotheses

Response

The following text has been added in the last sentence in the. Introduction in accordance with the
comment.

The null hypothesis of this study was that there was no difference in activities in the cervical
extensors and flexors in healthy adults and NSNP patients.

Methods:

Present key elements of study design early in the paper

Response

The following text has been added in the first paragraph of the 2. Methods in accordance with the
comment.

The procedures for the measurement and analysis of cervical muscle activity and as-sessment of
head and neck posture were performed according to previous study.

Clearly describe the diagnostic criteria for NSNP.

Response

Thank you for your comment. The NSNP, as the word nonspecific indicates, refers to a condition
that excludes specific diseases. Therefore, we stated the Exclusion Criteria to exclude specific
diseases.

Describe the setting, locations, and relevant dates, including periods of recruitment and data

Round 2

Reviewer 1 Report

Medicina-2009423-v2

Review: Characteristics of Activity of Cervical Extensors and Flexors in Nonspecific Neck Pain Patients: A Cross-Sectional Study

Proposed: major revision

Dear editor,

The authors has revised their paper very well, however, linguistically, the revision is poor. Before publishing this has to be revised.

 Additionally, there are still some shortcomings in this manuscript:

 Introduction:

Thank you for adding the correlation-part to the introduction. However, from a methodological stand point you are looking for associations tested by correlation. So please revise.

 Methods:

At the start of the study the authors now refer to their former study, however, a reference of this study is missing.

 Now you are using non-parametric tests, however, it is not clear if you tested whether your data were normally distributed. So please add at first how you tested for normality and then what you use when data are normally distributed and what you use when data were non-normally distributed.

 Results:

Don’t start a sentence with a number. For example 24 participants should be written as Twenty-four participants.

 Thank you for adding Table 1, however, these groups are still not statistically compared. Please do so.

To describe your results, let the reader know which data were not normal distributed.

Author Response

Responses to Reviewer 1 comments

Dear the reviewer:

Thank you for taking time to the second review and for the careful check and comments to improve our revised version. We have made revisions according to each comment. We hope our revisions are satisfactory this time.

Here are our answers.

Introduction:

Thank you for adding the correlation-part to the introduction. However, from a methodological stand point you are looking for associations tested by correlation. So please revise.

Response

According to the comment, we revised that sentence in the Introduction as below.

Also, we observed the craniovertebral angle (CVA) in NSNP patients to examine whether CVAs were associated with activities of the cervical extensors in the neutral position by using the flexor-extensor ratio to explore the pathogenesis of NSNP from the perspective of poor posture.

Methods:

At the start of the study the authors now refer to their former study, however, a reference of this study is missing.

Response

According to the comment, we added the reference number 25 in the Methods.

Now you are using non-parametric tests, however, it is not clear if you tested whether your data were normally distributed. So please add at first how you tested for normality and then what you use when data are normally distributed and what you use when data were non-normally distributed.

Response

According to the comment, we added the following in 2.6. Statistical Analysis.

A Kolmogorov-Smirnov normality test showed that ratios of muscle activity of the motions to the neutral position in 1/18 phases in the reference, 7/18 phases in control and 1/18 in NSNP group for the extensors and 1/18 in the control and 1/18 in NSNP group for the flexors: and for flexor–extensor ratios, in 6/21 in the reference and 4/21 phases in control group were normally distributed, but the remaining phases were not normally distributed. Therefore, we performed a nonparametric analysis and supplemented it with a parametric analysis.

Also, we added the following details about normality to the legend in Figure 2 and 3, respectively.

In the Figure 2

For the extensors, the ratios in Phase III in ipsilateral lateral flexion in the reference group, all phases in flexion, Phase I in extension, Phase I in contralateral lateral flexion, Phase II in ipsilateral rotation and Phase III in contralateral rotation in the control group, and Phase III in flexion in the NSNP group were normally distributed but not for the remaining phases. For the flexors, the ratios in Phase II in contralateral rotation in the control group, and Phase III in ipsilateral rotation in the NSNP group were normally distributed but not for the remaining phases.

In the Figure 3

The ratios in Phase III in extension, Phase II in contralateral lateral flexion, Phase II and III in ipsilateral rotation, Phase I and II in contralateral rotation in the reference group, and Phase I in extension, Phase I in ipsilateral rotation and Phase I and II in contralateral rotation in the control group were normally distributed but not for the remaining phases.

Results:

Don’t start a sentence with a number. For example 24 participants should be written as Twenty-four participants.

Response

According to the comment, we revised a number in the Results in question.

Thank you for adding Table 1, however, these groups are still not statistically compared. Please do so.

Response

We stated statistical comparisons between three groups in the 3.1 Demographics in the previous revised version as follows: “Demographic characteristic data of participants in the NSNP, control and reference group are shown in Table 1. The age in the NSNP group was significantly greater than the control (p < 0.001) and reference (p < 0.001) groups. The height in the NSNP group was significantly lesser than reference (p < 0.001) groups. The weight in the NSNP group was significantly lesser than the control (p = 0.031) and reference (p = 0.001) groups. There was no significant difference in Body Mass Index (BMI) among the three groups (p = 0.092).”

According to the comment, however, we put these results of statistical comparisons into the Table 1 and removed this part of the text.

To describe your results, let the reader know which data were not normal distributed.

Response

According to the comment, we added texts in question as above-mentioned.

Reviewer 2 Report

The authors have addressed all the comments raised by me. The manuscript can be accepted for publication.

Author Response

Dear the reviewer:

Thank you for the previous comments and reviewing the revised version.

The original manuscript has been much improved due to your comments.

Sincerely,

Authors